# Qualitative hepatitis C virus RNA assay identifies active infection with sufficient viral load for treatment among Phetchabun residents in Thailand

**Pornpitra Pratedrat[1], Pornjarim Nilyanimit[1], Rujipat Wasitthankasem[2], Nawarat Posuwan[3], Chompoonut Auphimai[1], Payuda Hansoongnern[1], Napaporn Pimsing[4], Saranya Ngamnimit[4], Chaiwat Thongmai[5], Wijittra Phaengkha[6], Nasamon Wanlapakorn[1], Sompong Vongpunsawad[1], Yong Poovorawan** [1,7] *

1 Center of Excellence in Clinical Virology, Faculty of Medicine, Chulalongkorn University, Bangkok, Thailand, 2 National Biobank of Thailand, National Science and Technology Development Agency, Pathum Thani, Thailand, 3 Chulabhorn International College of Medicine, Thammasat University, Rangsit Campus, Pathum Thani, Thailand, 4 Non-Communicable Disease Control Group, Phetchabun Provincial Health Office, Phetchabun, Thailand, 5 Phetchabun Provincial Public Health Office, Phetchabun, Thailand, 6 Namnao Hospital, Phetchabun, Thailand, 7 FRS(T), The Royal Society of Thailand, Bangkok, Thailand

* yong.p@chula.ac.th

**Data Availability Statement:** All relevant data are within the manuscript.

## Abstract

The World Health Organization envisions the elimination of viral hepatitis by 2030 through reducing prevalence and transmission, increasing diagnostic screening, and expanding treatment coverage. Efforts to micro-eliminate hepatitis in Phetchabun province in Thailand, a region where the prevalence of hepatitis C virus (HCV) infection and liver cancer is higher than elsewhere in the country, began with evaluating the province-wide burden of HCV. Here, we describe a feasibility study to assess active HCV infection by screening Phetchabun residents ages 35 to 69 years for anti-HCV antibodies by using a rapid diagnostic test (RDT) at the point of care. Positive anti-HCV results were further evaluated for active infection using qualitative HCV RNA assay, followed by quantitative HCV viral load determination in a subset of samples. Currently, we have identified 6.2% (10,621/170,163) anti-HCV positive individuals, of whom 74.9% (3,930/5,246) demonstrated detectable viral RNA. Quantitative test found that 97.5% (1,001/1,027) had HCV viral load ≥5,000 IU/mL. Thus, primary screening with anti-HCV RDT followed by qualitative HCV RNA evaluation could identify active and chronic HCV infection in almost all individuals with a viral load ≥5,000 IU/mL, which is the current threshold for treatment dictated by Thailand's National Health Security Office. Our data suggest that qualitative HCV RNA evaluation may obviate the need for the more expensive quantitative HCV viral load test and reduce a significant barrier toward HCV elimination in a middle-income country.

**Funding:** This study was supported by the National Research Council, Thailand Grand Challenge Fund (RES_64_058_30_020), Grant for New Researcher (NRCT5-TRG630014-04), and the Center of Excellence in Clinical Virology at Chulalongkorn University and Hospital. The Second Century Fund of Chulalongkorn University supported the postdoctoral fellowship position of Dr. Pornpitra Pratedrat. The funders had no role in study design, data collection and analysis, decision to publish, or preparation of the manuscript.

**Competing interests:** The authors have declared that no competing interests exist.

# Introduction

Globally, the hepatitis C virus (HCV) is a significant blood-borne pathogen, with chronic infection being the leading cause of liver cirrhosis and hepatocellular carcinoma (HCC) [1, 2]. Approximately 58 million people are chronically infected worldwide, with 1.5 million new HCV infections each year [3]. Furthermore, it is estimated that HCV contributed to >290,000 deaths in 2019. Most HCV infection remains persistent but asymptomatic throughout a person's lifetime, hindering early detection and treatment [4–6]. Direct-acting antiviral therapy is highly effective in eliminating infection in most patients regardless of viral genotype [7–10]. However, identifying active HCV infection and accessing treatment remain significant barriers in many underdeveloped regions.

In 2016, The World Health Organization Global Health Sector Strategy established the goal of viral hepatitis elimination by 2030. Targets include achieving 90% reduction in new infection, diagnosing 90% of currently infected patients, and ensuring treatment coverage for 80% of the chronically infected [11, 12]. However, success in the global viral hepatitis elimination rests on the commitment of governments to define, establish, and support national hepatitis treatment guidelines to ensure affordable care and provide treatment options for all HCV-infected patients. Countries considered on track to achieve hepatitis elimination include Australia, France, Iceland, Italy, Japan, South Korea, Spain, Switzerland, and the United Kingdom, all of which are high-income nations [13]. Less affluent countries such as Egypt, Georgia, and Mongolia are also expected to attain viral hepatitis elimination.

Although Thailand has a low national HCV seroprevalence (approximately 1%), Phetchabun province reportedly has a higher rate of pre-existing HCV infection than the rest of the country [14–16]. Although the overall anti-HCV prevalence there is 6.9% according to the latest study, the rate can vary in the north (10.8%) and south (2.9%) [15]. Due to its substantial HCV burden, an HCV micro-elimination feasibility study was undertaken in Phetchabun. However, a major obstacle to eliminating HCV in Thailand is the onerous preconditions necessary to access government-subsidized HCV treatment. In addition to positive antibody status, additional requirements include quantitative HCV RNA viral load ≥5,000 IU/mL and liver elastography or hepatic marker panel assay evaluation by a physician specialist. These out-of-pocket costs often deter patients with active hepatitis C from seeking timely medical attention and treatment.

Our previous study has shown that the anti-HCV rapid diagnostic test (RDT) is potentially helpful and reliable in identifying active HCV infection among Phetchabun residents [17]. In this study, we describe the province-wide effort to screen for anti-HCV in Phetchabun by examining the predictive usefulness of qualitative HCV RNA assay in identifying HCV patients with ≥5,000 IU/mL viral load. This work has implications for simplifying HCV screening, which could obviate the need for more expensive quantitative HCV RNA testing.

# Materials and methods

## Population

Phetchabun province has around 982,000 residents and is located approximately 300 km north of Bangkok. This province is 75 kilometers from east to west, 200 kilometers from north to south, and is administratively subdivided into 11 districts (https://pnb.hdc.moph.go.th/). Census data as of December 2020 indicate that there were 323,672 people ages 35 to 69 years who served as our study population. Most Phetchabun residents are employed in agriculture-related fields [15, 16]. The anti-HCV screening took place from September 2020 to December 2021, while HCV RNA testing was done in 2021 (April to December). Excluded among these

samples were HCV patients who had previously been treated and achieved a sustained virological response, individuals who self-reported to have chronic diseases, or who declined to screen. The Institutional Review Board of The Faculty of Medicine of Chulalongkorn University approved this study (IRB Number 028/63).

## Study design

Healthcare staff at the nearest local primary care centers or hospitals in Phetchabun interviewed and performed anti-HCV screening using RDT after the objective of the study was explained to participants. Only individuals who verbally consent were screened. Individuals with positive RDT results subsequently provided written informed consent for a confirmatory qualitative HCV RNA test involving venous blood collection by nurses at one of the three hospital hubs (Phetchabun Provincial Hospital, Lom Sak Hospital, and Wichian Buri Hospital) (Table 1), and quantitative HCV RNA viral load test on a subset of samples performed at a specialized laboratory in Bangkok.

## RDT for anti-HCV

As per manufacturer's instructions, finger-pricked whole blood (~10 μl) was mixed with the kit solution and tested with the Anti-HCV RDT (SD Bioline HCV, Abbott, North Chicago, IL). This RDT is a WHO pre-qualified rapid chromatographic immunoassay for the qualitative HCV evaluation within 5–20 minutes.

## Qualitative HCV RNA assay

Residents who tested positive for anti-HCV were referred to one of the three regional hub hospitals where blood plasma was collected. Staff at these hospitals performed the qualitative HCV RNA assay by using the COBAS AmpliPrep/COBAS TaqMan HCV Test, v2.0 implemented on the Cobas p 630 instruments according to the manufacturer's instructions (Roche Molecular Systems, Pleasanton, CA). This automated assay detects HCV genotypes 1–6 and reports that samples are either reactive ($\geq$15 IU/mL) or non-reactive (<15 IU/mL) (range 15 to $10^8$ IU/mL). The remaining plasma samples were transported frozen to Bangkok for quantitative viral load testing.

## Quantitative HCV RNA viral load test

Automated HCV detection involved the extraction of 200 μl plasma and RT-PCR on the Cobas 4800 System according to the manufacturer's instructions (Roche Diagnostics,

**Table 1. Regional hub and district hospitals in Phetchabun involved in the RDT and qualitative HCV RNA testing.**

| Region | Hospital Hub | District Area | District Hospital |
|---|---|---|---|
| North | Lom Sak Hospital | Lom Sak | Lom Sak Hospital |
| | | Lom Kao | Lom Kao Crown Prince Hospital |
| | | Nam Nao | Nam Nao Hospital |
| Central | Phetchabun Hospital | Mueang | Phetchabun Hospital |
| | | Chon Daen | Chon Daen Hospital |
| | | Khao Kho | Khao Kho Hospital |
| | | Wang Pong | Wang Pong Hospital |
| | | Nong Phai | Nong Phai Hospital |
| South | Wichian Buri Hospital | Wichian Buri | Wichian Buri Hospital |
| | | Bueng Sam Phan | Bueng Sam Phan Hospital |
| | | Si Thep | Si Thep Hospital |

Mannheim, Germany). This technology platform also detects HCV genotypes 1–6 and has a limit of detection (LOD) of 15.3 IU/mL (linear range 25 to $10^8$ IU/mL).

## Data analysis

Due to the global pandemic caused by the coronavirus disease 2019 (COVID-19), completion of the qualitative HCV RNA assay was disrupted at the provincial hospitals during this study. Only a subset of RDT-positive samples was promptly tested for viral RNA. Available results were then analyzed. Government-subsidized treatment for hepatitis C requires patients to possess HCV viral load of $\geq$5,000 IU/mL. Therefore, this value was used for subsequent comparison between qualitative and quantitative testing. We also compared differences in implementing an HCV micro-elimination pilot in our study versus that of the National Health Security Office (NHSO).

## Results

### Screening for anti-HCV using RDT

The prevalence of anti-HCV in 170,163 Phetchabun residents screened with RDT represents 52.6% coverage of the eventual target population (Table 2). Of these, 10,621 tested positive for RDT (6.2%). The top three districts with the most reported RDT-positive samples were from the north of the province, Lom Sak (12.2%), Lom Kao (11.2%), and Nam Nao (8.8%), while the fewest RDT-positive results were in Bueng Sam Phan and Si Thep (both 1.7%).

### Qualitative HCV RNA assay at three regional hospitals in Phetchabun

Since qualitative HCV RNA assay was more readily available and affordable when screening large numbers of individuals, it was used to preliminarily confirm the initial RDT results. Data from further testing of 5,246 out of 10,621 RDT-positive samples were available, of which 3,930 (74.9%) were HCV RNA-reactive (Table 2).

### Quantitative HCV viral load test in Bangkok

Among the 3,930 reactive samples, 1,027 were further tested for HCV RNA viral load. Overwhelming majority of the RNA-positive samples (97.5%, 1,001/1,027) demonstrated viral loads

**Table 2. HCV screening and confirmatory tests in 11 districts of Phetchabun.**

| District | RDT | | | Qualitative Test | | | Quantitative Viral Load (IU/mL) | | |
|---|---|---|---|---|---|---|---|---|---|
| | Tested | Positive | % | Tested | Reactive | % | Tested | $\geq$5,000 (%) | <5,000 (%) |
| Wang Pong | 7,104 | 268 | 3.8 | 69 | 57 | 82.6 | 38 | 35 (92) | 3 (8) |
| Nong Phai | 21,653 | 1,367 | 6.3 | 342 | 272 | 79.5 | 45 | 45 (100) | 0 |
| Chon Daen | 14,033 | 339 | 2.4 | 180 | 115 | 63.9 | 45 | 45 (100) | 0 |
| Lom Sak | 32,840 | 4,008 | 12.2 | 1964 | 1557 | 79.3 | 240 | 234 (98) | 6 (2) |
| Nam Nao | 5,226 | 458 | 8.8 | 208 | 140 | 67.3 | 140 | 137 (98) | 3 (2) |
| Bueng Sam Phan | 17,161 | 291 | 1.7 | 147 | 124 | 84.4 | 9 | 9 (100) | 0 |
| Lom Kao | 13,722 | 1,530 | 11.2 | 835 | 638 | 76.4 | 230 | 226 (98) | 4 (2) |
| Wichian Buri | 22,213 | 617 | 2.8 | 396 | 299 | 75.5 | 80 | 80 (100) | 0 |
| Si Thep | 8,833 | 152 | 1.7 | 129 | 108 | 83.7 | 79 | 73 (92) | 6 (8) |
| Khao Kho | 3,581 | 250 | 7 | 118 | 95 | 80.5 | 22 | 22 (100) | 0 |
| Mueang | 23,797 | 1,341 | 5.6 | 858 | 525 | 61.2 | 99 | 95 (96) | 4 (4) |
| **Total** | **170,163** | **10,621** | **6.2** | **5,246** | **3,930** | **74.9** | **1,027** | **1,001 (97.5)** | **26 (2.5)** |

≥5,000 IU/mL. In 19 other samples (1.9%, 19/1,027), viral loads were <5,000 IU/mL. There was no detectable HCV RNA in the remaining 7 samples (0.7%), possibly due to the higher threshold limit of detection for this test than the qualitative assay.

## Discussion

Greater proactive screening for past HCV exposure and follow-up evaluation to determine possible active infection is crucial to eliminating HCV and viral hepatitis. RDT use at the point of care combined with a non-quantitative HCV RNA assessment at the hospital is an alternative approach to identifying HCV-infected Thais. Still, this strategy currently lacks evidence-based data to support its implementation. The objective of this study was to conduct a feasibility study for the entire at-risk adult population of a Thai province with reportedly high HCV prevalence. We, therefore, undertook a province-wide screening for anti-HCV and evaluated the comparative usefulness of qualitative HCV RNA assay versus quantitative HCV viral load test. Despite the extenuating circumstances due to the COVID-19 pandemic, approximately half of the target Phetchabun residents underwent RDT. By screening for anti-HCV using a relatively affordable RDT locally and performing qualitative HCV RNA tests at 3 designated Phetchabun regional hospitals, we showed that, on average, 6.2% of the population ages 35 to 69 possessed anti-HCV. This age group comprises working-age adults, and defining HCV burden in this cohort has provided much clarity in understanding HCV in Thailand for this target group [14]. This prevalence in Phetchabun is much higher than the overall national seroprevalence of 0.9% in 2014 [14]. However, this rate is consistent with findings from our previous study conducted just several years ago on a much smaller Phetchabun population relative to the current study, which reported 6.4% positive rate by RDT and 6.9% by an automated chemiluminescence microparticle assay from surveying 4,769 residents [17].

Similar to our past observation, seroprevalence among districts varied by as much as 7 folds between northern and southern districts. However, the difference in seroprevalence cannot be explained by population number because the overall number of residents is similar for both regions. Other factors may be responsible for such observation, including how common tattooing and self-injecting recreational drugs are in each district [18]. These risk factors for HCV infection are well-established in studies involving culturally different patient groups in the U.S. [18, 19] and Australians [20]. Higher HCV infection is also associated with less education, agricultural occupation, and male gender in Thailand [15, 16].

The National Health Security Office (NHSO), operating under the Ministry of Public Health, provides and allocates an annual budget from the Thai government, which covers health insurance for Thais and sets the criteria for government-subsidized HCV treatment eligibility. Hence, the NHSO supports HCV screening, diagnostic, and drug coverage for HCV patients. The current policy for identifying active HCV infections eligible for subsidized treatment dictated by the NHSO guideline requires positive RDT screening and quantitative HCV RNA viral load with ≥5,000 IU/mL (Fig 1). Unfortunately, only Thais who present evidence of anti-HCV and HCV viral load ≥5,000 IU/mL can receive free treatment. However, the viral load test is not widely available throughout the country, but only at major hospitals. More importantly, a quantitative HCV RNA viral load test is unaffordable for most Thais because it is relatively expensive and payment is out-of-pocket. Consequently, HCV patients with <5,000 IU/mL viral load cannot receive free HCV treatment, most of whom are low-income earners. Many infected patients are therefore unable to access treatment. These patients, therefore, progress to cirrhosis and end-stage liver cancer as a result of the treatment barrier under the current NHSO policy. A study in 2019 reported that 52.2% to 62.5% of HCV infections were in an advanced stage of liver cirrhosis (F3 and F4). This data demonstrates that more than half

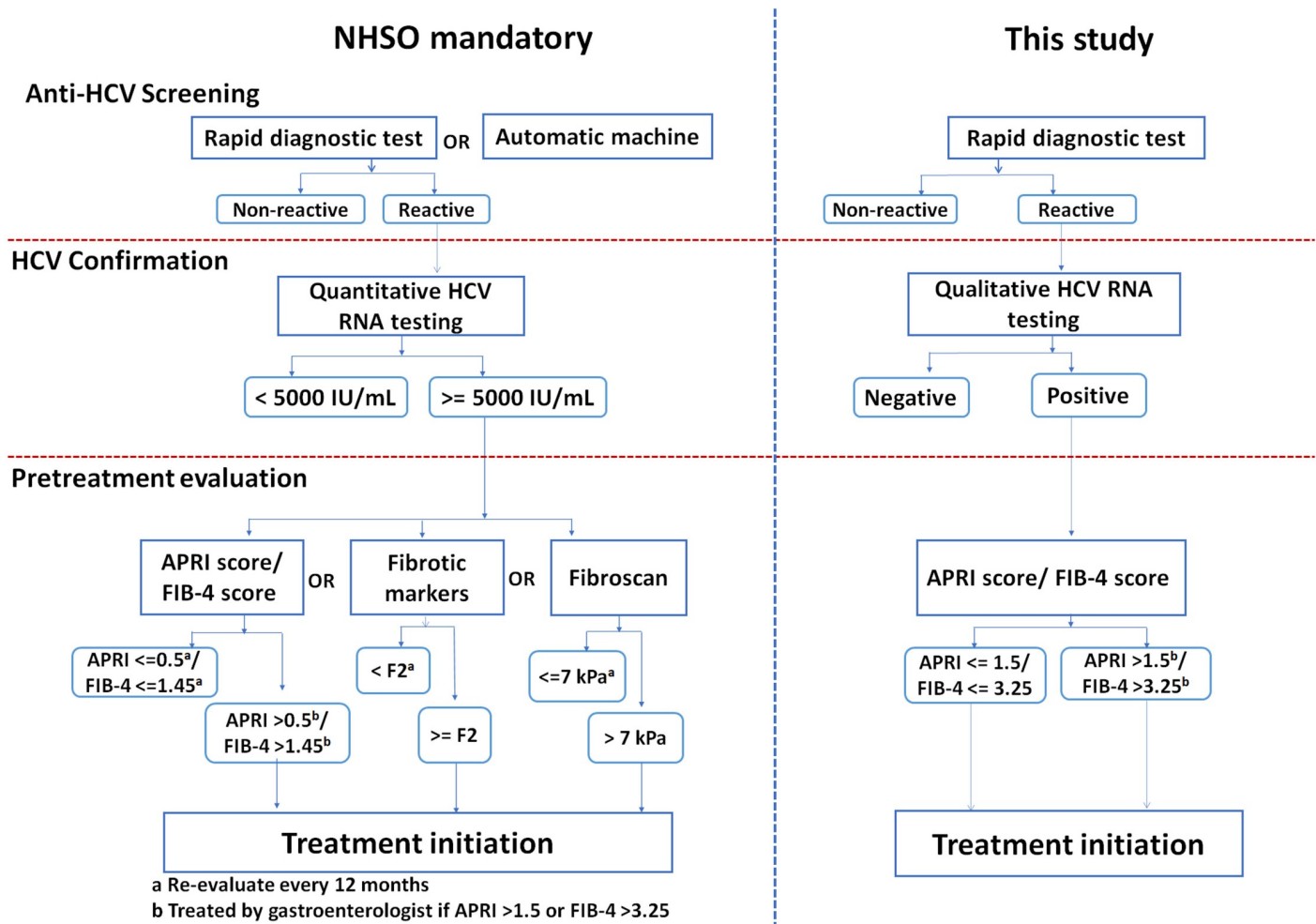

**Fig 1. Comparison of implementing a simplified HCV micro-elimination strategy supported by data from this study versus that of the National Health Security Office (NHSO).** Comparative anti-HCV screening, HCV confirmation and pretreatment evaluation for cirrhosis are shown.

of patients with HCV tend to develop liver cirrhosis and liver cancer if they do not receive any HCV treatment [21].

From testing half of the RDT-positive samples thus far, 75% have demonstrated active infection as determined by the qualitative HCV RNA test. Almost all were shown to possess viral load ≥5,000 IU/mL by the quantitative assay. The fact that the lower cost qualitative HCV RNA test (<11 USD each) can pinpoint 97.5% of patients with HCV viral load ≥5,000 IU/mL offers potentially significant savings and accessibility for HCV infection confirmation instead of the much more expensive quantitative viral load test, at least towards initiating treatment phase. Therefore, the qualitative HCV RNA is reliable and cost-effective for diagnosing active HCV infection, especially when nationwide implementation is considered. Our study provides evidence that qualitative HCV RNA highly correlated with HCV RNA ≥5,000 IU/mL, suggesting that viral load criteria could be eliminated. Even so, HCV viral load test remains valuable and essential for physicians to monitor HCV RNA viral load after the initiation of treatment.

This study has several limitations. Local staff did not record demographic information (name, age, gender, clinical signs) when administering RDT. Therefore, this information was

not available for analysis. Admittedly, screening could be done in only half of the targeted population throughout this study. Moreover, half of the RDT-positive results underwent qualitative HCV RNA tests, and only one-third of HCV RNA-reactive samples were subjected to quantitative HCV RNA viral load determination. Due to the extenuating circumstances as a result of the global COVID-19 pandemic for much of 2020 and 2021, patients and staff in Phetchabun are not immune to this disruption. Although test-to-treat strategy in high HCV prevalence setting is justifiably better than no intervention, middle-income countries may be hard-pressed to finance this strategy in low HCV prevalence setting, such as in areas with lower at-risk population or in affluent urban centers.

In conclusion, a revised practical guideline incorporating cost-effectiveness and a meaningful test-to-treat plan can help countries such as Thailand achieve HCV elimination by 2030. Our feasibility study in Phetchabun involving extensive population screening for active HCV infection can serve as an impetus toward an HCV micro-elimination effort in this province and subsequent expansion nationwide. The advent of pan-genotypic direct-acting antivirals has essentially rendered HCV genotype determination obsolete prior to treatment. By combining RDT and qualitative HCV RNA testing, another barrier to subsidized treatment qualification is removed and HCV diagnosis is expedited.

## Acknowledgments

We thank the staff of the Center of Excellence in Clinical Virology, Faculty of Medicine, Chulalongkorn University and King Chulalongkorn Memorial Hospital. We are grateful to the the healthcare staff at the sub-district health-promoting centers and all the hospitals, and especially the residents of Phetchabun who made this study possible.

## Author Contributions

**Conceptualization:** Napaporn Pimsing, Nasamon Wanlapakorn, Yong Poovorawan.

**Data curation:** Pornpitra Pratedrat.

**Formal analysis:** Pornpitra Pratedrat, Sompong Vongpunsawad.

**Funding acquisition:** Yong Poovorawan.

**Investigation:** Rujipat Wasitthankasem, Napaporn Pimsing, Saranya Ngamnimit, Chaiwat Thongmai, Wijittra Phaengkha, Yong Poovorawan.

**Methodology:** Pornpitra Pratedrat, Pornjarim Nilyanimit, Rujipat Wasitthankasem, Chompoonut Auphimai, Payuda Hansoongnern, Napaporn Pimsing, Saranya Ngamnimit, Yong Poovorawan.

**Project administration:** Rujipat Wasitthankasem, Nawarat Posuwan, Payuda Hansoongnern, Napaporn Pimsing, Saranya Ngamnimit, Chaiwat Thongmai, Wijittra Phaengkha, Yong Poovorawan.

**Resources:** Pornjarim Nilyanimit, Napaporn Pimsing, Yong Poovorawan.

**Supervision:** Rujipat Wasitthankasem, Wijittra Phaengkha, Yong Poovorawan.

**Validation:** Rujipat Wasitthankasem, Yong Poovorawan.

**Visualization:** Pornpitra Pratedrat, Rujipat Wasitthankasem, Sompong Vongpunsawad.

**Writing – original draft:** Pornpitra Pratedrat, Rujipat Wasitthankasem, Nasamon Wanlapakorn, Sompong Vongpunsawad, Yong Poovorawan.

**Writing – review & editing:** Rujipat Wasitthankasem, Sompong Vongpunsawad, Yong Poovorawan.

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
