## [Decision Letter · Decision Letter 0]

6 Sep 2022

PONE-D-22-13140Qualitative hepatitis C virus RNA assay identifies active infection with sufficient viral load for treatment among Phetchabun residents in ThailandPLOS ONE

Dear Dr. Poovorawan,

Thank you for submitting your manuscript to PLOS ONE. After careful consideration, we feel that it has merit but does not fully meet PLOS ONE’s publication criteria as it currently stands. Therefore, we invite you to submit a revised version of the manuscript that addresses the points raised during the review process.

Your manuscript was reviewed by one expert in the field. Unfortunately, other potential reviewers could not accept the invitation. The reviewer identified essential problems which require your attention. I would like to add that your work is important and may benefit from a more extensive discussion of cost-effectiveness of your strategy. Also, it would be of interest to readers to know your opinion on how the suggested testing strategy is applicable to low prevalence populations. Please consider discussing briefly whether separation of local anti-HCV and central HCV RNA testing may result in losses to care in low and high prevalence settings? It seems that your strategy of using qualitative HCV RNA assays may reduce such losses among high prevalence populations.  Will it work in general population with low level of HCV infections?

We look forward to receiving your revised manuscript.

Kind regards,

Yury E Khudyakov, PhD

Academic Editor

PLOS ONE

Journal Requirements:

a) Did participants provide their written or verbal informed consent to participate in this study?

Reviewers' comments:

Reviewer's Responses to Questions

**Comments to the Author**

1. Is the manuscript technically sound, and do the data support the conclusions?

Reviewer #1: Yes

2. Has the statistical analysis been performed appropriately and rigorously? 

Reviewer #1: Yes

3. Have the authors made all data underlying the findings in their manuscript fully available?

Reviewer #1: Yes

4. Is the manuscript presented in an intelligible fashion and written in standard English?

Reviewer #1: Yes

5. Review Comments to the Author

Reviewer #1: Thank you for asking me to review this study that confirms that almost three-quarters of patients testing RDT positive for hep C in a province in Thailand are viraemic. The study supports the use of a qualitative test and confirms the vast majority,~98% have HCV VLs >5000IU/ml. I have just a few comments/questions of clarity:

1. Could the authors explain the basis of Thai Government policy to only offer treatment to patients with HCV VLs >5000IU/ml?

2. Why did the authors consider using a qualitative test? Was this to confirm its predictive value? Is there a cost saving on the qualitative test? The qualitative test helps in confirming that most had VLs >5000IU/ml - there is no clarity on why the qualitative test was performed?

3. Were any patients identified in the study linked to care? If not, will this still occur?

6. PLOS authors have the option to publish the peer review history of their article (what does this mean?). If published, this will include your full peer review and any attached files.

Reviewer #1: No

---

## [Author Response · Author response to Decision Letter 0]

12 Oct 2022

Response to Editor’s Comments

I would like to add that your work is important and may benefit from a more extensive discussion of cost-effectiveness of your strategy. 

We agree with the Editor, but unfortunately it is not in our expertise to perform this calculation. Too many factors complicate an accurate calculation of cost-effectiveness, such as whether our proposed strategy will eventually be adopted by the Thai government, whether the latter will have sufficient buying power to negotiate affordable pricing for the tests, and the ever-increasing inflation rate, which this year is currently at 8%. 

Also, it would be of interest to readers to know your opinion on how the suggested testing strategy is applicable to low prevalence populations. 

We honestly do not know at what prevalence rate our strategy will be useful and applicable. In affluent countries, which have low prevalence of HCV infection, diagnostic and treatment are either covered by private health insurance or socialized government-run healthcare programs. There, cost-effectiveness is really not a factor in whether someone gets treated. Given that the national HCV seroprevalence in Thailand as a whole is low, our speculation on the applicability in the test-to-treat strategy in low prevalence setting may jeopardize our argument for an expanded HCV diagnostic/treatment. We hope the Editor will understand why we have refrained from addressing this in the manuscript. 

Please consider discussing briefly whether separation of local anti-HCV and central HCV RNA testing may result in losses to care in low and high prevalence settings? It seems that your strategy of using qualitative HCV RNA assays may reduce such losses among high prevalence populations. Will it work in general population with low level of HCV infections?

We can envision that testing in high prevalence setting is justifiable given the ultimate financial cost of doing nothing (treating patients when they already developed liver cancer). In low prevalence setting, middle-income countries may not be able to justify the financial resource required for micro-elimination. Again, we do not know the threshold of the prevalence of HCV infection for the strategy to be effective. We have added this in the study limitation in the Discussion: “Although test-to-treat strategy in high HCV prevalence setting is justifiably better than no intervention, middle-income countries may be hard-pressed to finance this strategy in low HCV prevalence setting, such as in areas with lower at-risk population or in affluent urban centers.”

Response to Reviewer’s Comments

1. Could the authors explain the basis of Thai Government policy to only offer treatment to patients with HCV VLs >5000IU/ml?

We are embarrassed to say that, after inquiring about this policy with officials in the Thailand Ministry of Public Health, no one seems to know. 

2. Why did the authors consider using a qualitative test? Was this to confirm its predictive value? Is there a cost saving on the qualitative test? The qualitative test helps in confirming that most had VLs >5000IU/ml - there is no clarity on why the qualitative test was performed?

Qualitative test costs one-fourth the price of the quantitative test, so it was a more feasible test to use. We have stated this in the revised Result section “Since qualitative HCV RNA assay was more readily available and affordable when screening large numbers of individuals, it was used to preliminarily confirm the initial RDT results”. Overall results from the qualitative test also helped to demonstrate its non-inferiority compared to the quantitative test.

3. Were any patients identified in the study linked to care? If not, will this still occur?

Yes, the patients identified in this study are linked to care. At the time preparing this study, the UC criteria has abandoned that viral load cut-off and allowed HCV treatment in all infected patients.

---

## [Editor Report · Decision Letter 1]

21 Oct 2022

Qualitative hepatitis C virus RNA assay identifies active infection with sufficient viral load for treatment among Phetchabun residents in Thailand

PONE-D-22-13140R1

Dear Dr. Poovorawan,

We’re pleased to inform you that your manuscript has been judged scientifically suitable for publication and will be formally accepted for publication once it meets all outstanding technical requirements.

Kind regards,

Yury E Khudyakov, PhD

Academic Editor

PLOS ONE
---

## [Editor Report · Acceptance letter]

6 Jan 2023

PONE-D-22-13140R1 

Qualitative hepatitis C virus RNA assay identifies active infection with sufficient viral load for treatment among Phetchabun residents in Thailand 

Dear Dr. Poovorawan:

I'm pleased to inform you that your manuscript has been deemed suitable for publication in PLOS ONE. Congratulations! Your manuscript is now with our production department. 

Kind regards, 

on behalf of

Dr. Yury E Khudyakov 

Academic Editor

PLOS ONE